# Short-Term Foreshocks as Key Information for Mainshock Timing and Rupture: The M_w_6.8 25 October 2018 Zakynthos Earthquake, Hellenic Subduction Zone

**DOI:** 10.3390/s20195681

**Published:** 2020-10-05

**Authors:** Gerassimos A. Papadopoulos, Apostolos Agalos, George Minadakis, Ioanna Triantafyllou, Pavlos Krassakis

**Affiliations:** 1International Society for the Prevention & Mitigation of Natural Hazards, 10681 Athens, Greece; agalosapostolos@gmail.com; 2Department of Bioinformatics, The Cyprus Institute of Neurology & Genetics, 6 International Airport Avenue, Nicosia 2370, P.O. Box 23462, Nicosia 1683, Cyprus; g.minadakis@gmail.com; 3The Cyprus School of Molecular Medicine, The Cyprus Institute of Neurology & Genetics, 6 International Airport Avenue, Nicosia 2370, P.O. Box 23462, Nicosia 1683, Cyprus; 4Department of Geology & Geoenvironment, National & Kapodistrian University of Athens, 15784 Athens, Greece; itriantaf@geol.uoa.gr; 5Centre for Research and Technology, Hellas (CERTH), 52 Egialias Street, 15125 Athens, Greece; p_krassakis@hotmail.gr

**Keywords:** seismicity cluster, swarm, foreshocks, precursory seismicity, predictive value, Hellenic Subduction Zone, 2018 Zakynthos earthquake, rupture process

## Abstract

Significant seismicity anomalies preceded the 25 October 2018 mainshock (M_w_ = 6.8), NW Hellenic Arc: a transient intermediate-term (~2 yrs) swarm and a short-term (last 6 months) cluster with typical time-size-space foreshock patterns: activity increase, b-value drop, foreshocks move towards mainshock epicenter. The anomalies were identified with both a standard earthquake catalogue and a catalogue relocated with the Non-Linear Location (NLLoc) algorithm. Teleseismic P-waveforms inversion showed oblique-slip rupture with strike 10°, dip 24°, length ~70 km, faulting depth ~24 km, velocity 3.2 km/s, duration 18 s, slip 1.8 m within the asperity, seismic moment 2.0 × 10^26^ dyne*cm. The two largest imminent foreshocks (M_w_ = 4.1, M_w_ = 4.8) occurred very close to the mainshock hypocenter. The asperity bounded up-dip by the foreshocks area and at the north by the foreshocks/swarm area. The accelerated foreshocks very likely promoted slip accumulation contributing to unlocking the asperity and breaking with the mainshock. The rupture initially propagated northwards, but after 6 s stopped at the north bound and turned southwards. Most early aftershocks concentrated in the foreshocks/swarm area. This distribution was controlled not only by stress transfer from the mainshock but also by pre-existing stress. In the frame of a program for regular monitoring and near real-time identification of seismicity anomalies, foreshock patterns would be detectable at least three months prior the mainshock, thus demonstrating the significant predictive value of foreshocks.

## 1. Introduction

Significant increases of seismic activity in space and time, termed seismicity clusters, were recognized long ago. Pioneering research performed in the 1960s described three main types of clusters: foreshocks, swarms, aftershocks [1,2]. Foreshocks precede a forthcoming stronger earthquake in the same area and in a short time, e.g., in a few days or up to a few months [3,4,5,6,7,8]. Therefore, such a type of cluster is usually termed short-term foreshock sequence. In the remaining part of this paper, the term “foreshocks” is meant as “short-term foreshocks” unless otherwise indicated.

Although several mechanisms have been proposed for the foreshock generation, e.g., [9,10,11,12,13], they are considered as a promising precursor for the prediction of mainshocks, e.g., [14,15,16]. Foreshocks may precede not only strong and large earthquakes but also small-to-moderate ones [3,17,18,19,20]. However, space-time clusters, which do not conclude with the generation of a stronger earthquake exceeding significantly the size of the rest cluster events, are transient seismicity swarms. Aftershock activity follows a mainshock, but the discrimination between foreshocks and other types of clusters is a puzzling issue [21,22,23]. On the other hand, strong or large earthquakes may happen without precursory seismicity anomalies. In some instances, however, this may be only an artificial result since foreshocks escape recognition from routine procedures of seismic analysis in conditions of low monitoring capabilities, e.g., [24]. Although foreshock incidence depends on several factors, including crustal heterogeneity [25,26], focal mechanism and tectonic style [17,27], the recognition of foreshock rate progressively increased in the last century or so from about 20% [28,29] to 50% [30] and even to 70% [8]. These results showed that short-term foreshock occurrence in nature is more prevalent than previously thought [8]. However, lower foreshock rates have been reported, e.g., [31,32], but usually this depends on foreshock definition.

Since no standard methods have been established so far to discriminate between foreshocks and other seismicity clusters at least in real time conditions, the *a posteriori* examination of case studies is of particular importance for better understanding geophysical and statistical features that may support discrimination of the various types of seismicity clusters, highlight the predictive value of foreshocks and reveal the foreshocks role in the mainshock rupture process. Proposed discriminants of various seismicity cluster types include statistical space-time-size patterns [16,19,21,22,23], waveform similarity [18,25,33] and changes in focal mechanism and stress field [34,35].

In this study, we examine statistically significant seismicity anomalies prior to the strong mainshock (moment magnitude M_w_ = 6.8) that ruptured the Hellenic Subduction Zone northwest section, south of Zakynthos Isl. in the Ionian Sea, on 25 October 2018 (Figure 1). We outline the seismotectonic setting of the study area and investigate premonitory seismicity patterns through statistical analysis and testing for significance. We show that a transient intermediate-term (1.5–2 yrs) swarm of low magnitude events occurred prior the mainshock. After a pause of the seismic cluster, accelerated short-term foreshocks occurring in the last six months culminated in the mainshock rupture. Initially we used a standard non-relocated earthquake catalogue aiming to understand the foreshocks predictive power in nearly real-time conditions. Then, we relocated the catalogue, verified the seismicity anomalies, developed a finite-fault model by inverting teleseismic mainshock P-waveforms, and eventually correlated the premonitory seismic anomalies with the mainshock nucleation and rupture process, as well as with the early aftershocks spatial distribution based on Coulomb stress transfer modeling.

## 2. Materials and Methods

### 2.1. Seismotectonic Setting

#### 2.1.1. The Hellenic Subduction Zone (HSZ)

The HSZ is the most active geodynamic structure in the Mediterranean region due to the subduction of the Nubian (African) oceanic lithosphere beneath the Aegean region at rates of about 35–40 mm/yr [36,37,38]. The lithospheric subduction is the main geodynamic process that controls the HSZ seismicity which takes its highest level in the western HSZ segment, e.g., [39], where the upper and lower plate convergence is nearly normal to the plate boundary (Figure 1a) [40,41,42,43,44,45]. The relative motion between the Aegean region and the subducting slab is accommodated by shallowly dipping fault planes with dip angles <20° along the interplate seismogenic zone as indicated by fault-plane solutions [41,42,43,44,45]. Higher-angle reverse faults (dip angles > 30°), likely splaying off from the plate interface, are observed at shallower depths with respect to the interplate earthquakes [45,46] and have been observed as playing a major role in the seismogenesis along the outer Hellenic fore-arc, e.g., [47,48]. The predominant compressive stress regime observed in the study area to the west and southwest of Peloponnese changes to strike-slip in the area to the NW of Zakynthos, Kefalonia and Lefkada Islands along the so-called Kefalonia Fault Zone, e.g., [49] (Figure 1a).

#### 2.1.2. The 2018 Earthquake

On 25 October 2018, 22:54:51 UTC, a strong earthquake (M_w_ = 6.8) ruptured the HSZ western section to the south of Zakynthos Isl., Greece (Figure 1b). Fault-plane solutions produced by various centers (see summary in https://www.emsc-csem.org/Earthquake/mtfull.php?id=720235&year= 2018;INFO) revealed thrust movement combined with a significant strike-slip component (Figure 1b). Τhe rupture process of the earthquake was approached by several research teams, based mainly on the inversion of near-field strong-motion and teleseismic waveform records, GPS displacements and tsunami records. However, no consistent results were obtained. One approach [50] indicated that the inverted slip distribution is characterized by two asperities with peak slip of 1.4 m at depth 17 km. The rupture propagated unilaterally but mainly towards N-NE and with a down-dip extent up to 38 km, indicating that this event broke the entire underthrust Ionian oceanic crust beneath the megathrust. Others [51] supported that the earthquake ruptured the Hellenic megathrust plate interface along a N-S striking thrust/oblique-slip fault with a length of ~26 km and low-dip angle (23°) but at a depth of <15 km. On the other hand, a complex pattern was suggested for the 2018 mainshock consisting of two fault segments: a low-dip thrust, and a dominant, moderate-dip, right-lateral strike-slip, both in the crust [52]. Slip vectors, oriented to SW, are consistent with the plate motion and the sequence can be explained in terms of trench-orthogonal fractures in the subducting plate and reactivated faults in the upper plate [52]. According to a different analysis [53], the preferred rupture model is the one with a N9°E-striking 39°ESE-dipping plane. The rupture history involved a bilateral propagation, featuring two asperities: a main slip patch extending between 14 and 28 km in depth, 9 km NE from the nucleation, and a slightly shallower small patch located 27 km SW from the nucleation. The maximum slip was found at 1.8 m and the total seismic moment 2.4×10^19^ Nm, corresponding to M_w_ = 6.8. The slip angle shows a dominant right-lateral strike-slip mechanism, with a minor reverse component that increases on the deeper region of the fault [53].

### 2.2. Patterns of Clustered Seismicity

#### 2.2.1. Definitions

Terms like foreshocks, swarms and clusters often are used vaguely without appropriate definition, one term being used instead of the other. Such terms are frequently used to describe supposed seismicity anomalies without testing for their statistical significance. Such inappropriate use of the seismicity behavior puts in doubt models or interpretations that connect seismicity patterns with seismotectonics and foreshock generation mechanisms. Motivated by such disadvantages, we proceed with some definitions needed to follow the subsequent sections of the paper.

The various states of seismicity can be approached by three time-space-size attributes: the seismicity rate, r, i.e., the event count per time unit, the spatial distance between earthquake epicenters, D, and the parameter b, i.e., the slope of the straight line expressing the magnitude-frequency relation [54,55]
log *N* = *a* − *bM*,(1)

N is the cumulative number of events with magnitude ≥M; a, b are parameters determined by the seismicity data. Suppose a time interval extending from T_N_ to T_o_ and composed by several sub-intervals, e.g., T_b_ from T_N_ to t_i_, representing a background seismicity period, and T_f_ from t_i_ to T_o_, representing a foreshock sequence prior a stronger mainshock at time T_o_; i = 1, 2, 3…n (Figure 2). A background seismicity period is characterized by no statistically significant anomalies in the three attributes. Otherwise, statistically significant anomalies signify the presence of seismicity clusters, which deviate from the background activity. If a significant seismicity anomaly (cluster) occurring before T_o_ is not foreshock activity, then it would be a transient swarm which does not culminate in a significantly stronger mainshock. Aftershock sequences, occurring after T_o_ in Figure 2, are the most characteristic examples of seismicity clusters.

#### 2.2.2. Seismicity Patterns

Seismicity studies performed in several tectonic environments have shown that foreshocks increase towards the mainshock as the inverse of time and in a timeframe varying from hours to a few days (imminent foreshocks) up to 5-6 months (short-term foreshocks), e.g., [4,5,6,16,19,56,57,58,59,60]. The acceleration of the fracturing process is supported by laboratory material fracture experiments, e.g., [1,61,62], numerical modeling in spring-block models, e.g., [63,64,65] and analytical damage mechanics modeling, e.g., [66]. Interestingly, the seismicity often accelerates in a long-term sense as well, e.g., in a time window of several years before strong or large earthquakes [67,68,69], while in some instances gradual long-term drop of b-value has been also observed [70]. The so-called long-term foreshocks appear in a much wider area than the short-term ones. However, it has not been examined so far how the two processes of acceleration are possibly interrelated. In aftershock sequences, the activity decreases exponentially with time according to the Omori law and in a timeframe up to months or even years; see review in [71].

It is common knowledge that in foreshock sequences, b drops with respect to aftershocks and background seismicity in the same area [1,2,4,6,16,19,72,73,74,75,76,77,78]. Τhis implies that foreshock sequences are relatively enriched in higher magnitude events. The b-value depends on several factors including stress loading conditions and tectonic regime [1,61,79] as well as crustal heterogeneity, e.g., [26]. High b indicates low stress is asymmetrically distributed, while low b is an evidence of concentrated high stress. Within this concept it is explainable why increased b-value is observed as a rule in aftershock sequences and swarms. On the contrary, foreshock sequences are characterized by low b-value and can be modeled by a process of material softening in the seismogenic volume, e.g., [65]. Therefore, b is considered as a stress meter [79,80] which should be seen as a field variable, not as a constant [81].

In the space domain, it has been found that foreshocks tend to move towards the mainshock epicenter [16,19,27,60,82], providing evidence that foreshock incidence perhaps is an inherent property of the mainshock nucleation process. However, this property frequently is hidden due to large errors involved in the epicentral determinations. The aftershock area, however, gradually expands at least in the first days after the mainshock occurrence [83,84].

The time-size-space foreshock patterns described above were mainly established by either averaging over several foreshock sequences or analyzing synthetic earthquake catalogues. This is due to the scarcity of data since most seismicity catalogues usually are not of adequate accuracy and completeness level. Therefore, the co-existence of the three patterns in individual foreshock sequences has been verified only in a very limited number of cases so far. The 6 April 2009 L’Aquila (Central Italy) M_w_ = 6.3 earthquake is the first example of mainshock preceded by strong foreshock activity in the three domains of space, time and size [16]. Good examples were also found prior to mega earthquakes (M > 8) in the Chilean subduction zone [60], but also before small-to-moderate (M~5) earthquakes in an extensional tectonic environment in Greece [19]. Considering a critical area of radius R from the mainshock epicenter for the foreshock occurrence, we found that R~15 km in the L’Aquila earthquake, R > 50 km for three Chilean mega subduction earthquakes and R < 10 km for small-to-moderate Greek earthquakes [19]. Consequently, there is evidence that the mainshock magnitude likely scales with the foreshock area [19], although more research is needed for verification.

We consider a cluster as a short-term foreshock sequence under two conditions: (1) the seismicity rate, r, increases while the b-value decreases as compared to the respective values during the period of background seismicity in the same area; decrease of distance parameter, <D>, is an additional but not decisive criterion since such a drop may appear during transient swarms as well; (2) non-random changes are detected at least for r and b. We adopt that strong foreshock signal is evident when at least both these changes are highly significant at the level of 95%, while weak foreshock signal is adopted if at least one change is significant, i.e., ranging between the levels of 90% and 95%.

#### 2.2.3. Testing

To examine non-random seismicity changes, we tested a hypothesis which generally is stated as: “two non-overlapping time segments of an earthquake catalogue represent significantly different seismicity states”, e.g., T_b_ and T_f_ in Figure 2. The significance of r and <D> changes is controlled with the statistical z-test. The so-called Utsu test [84,85] is used to test changes of b. The probability, P, that two samples with different b-values come from the same population is
P ≈ exp [−(dA/2) −2],(2)
dA depends on the *b*-values and on the number of events inserted in the two earthquake samples [85], see Appendix B. Low P values, e.g., 0.05 or less, indicate high statistical significance in the change of b, implying that the two samples do not come from the same earthquake population.

### 2.3. Identification of Seismicity Anomalies

To identify seismicity anomalies, we performed analysis with the FORMA algorithm (FOReshock-Mainshock-Aftershock) developed in-house and detailed elsewhere [16,19,60]. In a given spatiotemporal window, the time variations of r, b and <D> as well as the non-randomness in their changes are examined. Seismicity rate, r, is determined with the least-squares regression. The maximum likelihood estimator of b is [86]
b = 1/ln(10)(<M> − M_C_),(3)
where M_c_ is the lower magnitude threshold in the earthquake sample determined from the data fit to formula (1). Since the determination of b is quite sensitive to several factors, an alternative is to examine the time variation of the mean magnitude, <M>. Therefore, in our analysis we examined changes of both b and <M>.

In such a type of analysis, a usual but arbitrary practice followed in many relevant studies is to examine seismicity in predefined spatiotemporal windows. Selection of narrow windows may result in missing possible foreshocks or other seismicity anomalies. On the other hand, selection of quite wide windows may contaminate seismicity in the target area with seismicity anomalies associated with faults or fault zones in the surrounding areas. To avoid such issues, we investigated an optimum window selection. We start with windows wide enough to detect possible contaminant seismicity and gradually narrowed down the windows until significant seismicity changes, if any, would be detected. FORMA performs tests for the identification of statistically significant changes of r and b for all the possible pairs of sequential seismicity intervals, e.g., T_b_ and T_f_ in Figure 2. In this way, the onset and termination of seismicity anomalies are detected along with their significance levels. The next step is the calculation of the mean Euclidean distance, <D>, between earthquake epicenters and the mainshock epicenter. Hypothesis testing follows as outlined in Section 2.2. Our approach resembles the unified scaling law describing seismicity changes in the domains of rate, energy and space [87,88]. However, we examined separately the three attributes, for the reason that we were interested to independently check the statistical significance of each one of changes in the three domains.

The maximum likelihood approximation (formula 3, see Ref. [86]), is the standard method that we followed to calculate b, hereafter noted as b_ML_. For reasons of comparison, we calculated also b_GR_ with the weighted least-square method to the magnitude-frequency distribution (Formula 1). Since b is sensitive to M_c_, it is calculated for various levels of M_c_. Another condition we adopted was δ = Μ_max_ − M_min_ > 1.4, since for δ<1.4 the calculation of b is unstable [89]; Μ_max_ and M_min_ are the maximum and minimum magnitudes inserted in the earthquake sample, respectively. The time variation of b is examined by applying a conditional backward windowing technique. More details on b-value computation can be found in Appendix B.

Implementing the methodology in the 25 October 2018 mainshock case, we initially analyzed the earthquake catalogue of the Institute of Geodynamics, National Observatory of Athens (NOA) (see Section 2.6 Data), for a time interval extending five years before the mainshock and at radius R = 150 km around its epicenter. The investigation easily detected contaminant seismicity, the most important being the aftershock sequence of the earthquakes of 26 January (M_w_ = 6.0) and 3 February (M_w_ = 5.9) 2014 [90] occurring at distance ~120 km from the 2018 mainshock epicenter (Figure 1a). Narrowing down the area, we found that a time variation of the seismicity rate, r, became gradually evident, e.g., for R~60 km (Figure 3). However, for radius R = 30 km, the change in r is stronger with two steady rate (background seismicity) periods alternating with periods of increased rate (Figure 4), as detailed in the results section.

The above observations imply that for R>30 km, the earthquake sample is enriched in background seismicity events and, therefore, the anomalies lose strength. On the other hand, for R < 30 km, the anomalies gradually weaken since the earthquake sample loses events. Therefore, we eventually analyzed the seismicity developed within radius of R = 30 km around the 25 October 2018 mainshock epicenter and within the time interval from 1 January 2014 up to the mainshock occurrence. The mainshock, however, has not been included in our statistical analysis to avoid biased calculation of b. The results obtained are presented in Section 3.

### 2.4. Seismicity Relocation

Relocation of the standard NOA catalogue initially used is of double value. First, it helps to check the robustness of the seismicity variation results obtained in Section 3.1. Second, relocated hypocenters help to better understand possible correlations between the premonitory seismicity anomalies and the mainshock rupture process and the aftershock spatial distribution. Therefore, we relocated the hypocenters of the 2018 mainshock, of the preceding seismicity as well as of the early aftershocks occurring in the first 5 days after the mainshock, i.e., until 30 October 2018. Later aftershock areas expand due to gradual inelastic stress redistribution and, therefore, they are not representative of the seismic rupture dimensions [83,84]. Relocation was performed by utilizing the phase onsets determined by NOA (see Data sub-section).

We employed the Non-Linear Location (NLLoc) algorithm [91], which follows the non-linear location approach [92], giving a complete, probabilistic solution of the earthquake location problem expressed in terms of the Posterior Density Function (PDF) in the space-time domain. The 1D velocity model adopted [93] provides a better distribution of the hypocenters as well as slightly better vertical errors and Root Mean Square (RMS) with respect to other models available for the study region, e.g., [94,95]. Only phases recorded at permanent stations of the Hellenic Unified Seismological Network situated within the epicentral distance of 120 km from the mainshock were utilized. The reasons were detailed analyzing the January–February 2014 earthquake sequence to the north of the study area [96] (Figure 1a). Following an iteration procedure, the relocation was repeated by including phase residuals at seismic stations obtained from the previous run until no further significant decrease of the RMS value between two successive runs was achieved. To correct for irregular station distribution, a relevant station weighting tool in NLLoc was utilized.

### 2.5. Finite-Fault Model

The space-time evolution of the mainshock rupture has been modeled by following a finite-fault methodology [97,98,99] based on the inversion of teleseismic P-waveforms (see Data sub-section). A kinematic parameterization of the earthquake fault was used to identify the space-time slip distribution. The synthetics were calculated for each cell in which the fault divided and compared with the recorded data during an inversion, producing the solution vector, i.e., the slip for each cell. The methodology has been successfully implemented in other strong Mediterranean earthquakes, e.g., [90,100], and, therefore, it is not further analyzed here.

With the relocation procedure, the 25 October 2018 mainshock hypocenter was placed at 37.388° N, 20.557° E and at depth of h = 12 km. This solution shifted slightly NOA’s determination (37.35° N, 20.49° E, h = 13 km) towards northeast (Figure 1b). Based on this solution, a rectangular fault was assumed to be 70 km in length with a depth range from 0 to 24 km. Such fault dimensions are large enough to describe successfully the slip for an earthquake as large as M_w_ = 6.8. The rake vector, however, was allowed to vary from 90° (pure thrust) to 180° (pure strike-slip) permitting faulting combining up-dip and right-lateral strike-slip movements of the hanging wall domain relatively to the footwall. The hypocenter was set at nearly equal horizontal distances from the south and north fault edges, respectively. This selection permitted the seismic rupture to propagate bilaterally if imposed by the waveform data. The fault was discretized to 108 sub-faults or cells, 18 of them along strike and 6 along dip, while several rupture velocity values, varying from 2.6 km/s to 3.6 km/s, were tested. The rupture velocity did not change during the inversion, but the fact that six time windows were inserted, using 1.0 s time lag, allowed a rise time of up to 6 s to be taken on each sub-fault, if required by the observations.

### 2.6. Data

Two earthquake catalogue datasets have been used both for the period from 1 January 2014 to 30 October 2018. The first set has been retrieved from the standard catalogue of NOA (http://www.gein.noa.gr/en/seismicity/earthquake-catalogs). The second dataset is a catalogue produced with the relocation procedure followed in this study by utilizing the phase onsets determined by NOA (https://bbnet.gein.noa.gr/HL/databases/database). A uniform uncertainty of 0.5 s has been associated to all pick quality classes.

For the seismic rupture model, the dataset consisted of P-waveforms from 30 station records at teleseismic distances ranging from 30° to 90° with good azimuthal coverage (Figure 5). The waveform data were downloaded from the Incorporated Research Institutions for Seismology Data Management Center (https://ds.iris.edu/ds/nodes/dmc/). The records were deconvolved to ground displacement, bandpass-filtered (0.03–1.0 Hz) and resampled to 0.2 s common time step. The fit between real waveforms and synthetics is shown in Figure 6.

## 3. Results

### 3.1. Seismicity Changes

Seismicity analysis performed for the entire time interval from 1 January 2014 until the 2018 strong earthquake occurrence and within a radius of R = 30 km around its epicenter shows that four states of seismicity can be distinguished (Figure 4). The first (1 January 2014–1 August 2016) is characterized by steady seismicity rate, r, and represents background seismicity state (BGS 1) since no significant changes were noted in that period. This state was interrupted by an increase in r in the interval extended from 1 August 2016 to 1 May 2017 (cluster 1). Only small earthquakes of M_L_ < 4 occurred during cluster 1; M_L_ is local magnitude. A second state of stable r followed and lasted for about one year (BGS 2). However, increased rate resumed from 20 April 2018 until the 25 October 2018 mainshock (cluster 2). All the events remained small (M_L_ < 4) except the two largest imminent events that happened 8 days (M_L_ = 4.2, M_w_ = 4.1) and 50 min (M_L_ = 4.9, M_w_ = 4.8) prior the mainshock (Figure 1b and Figure 4).

### 3.2. Hypotheses Testing

Based on the seismicity variations found, we tested two hypotheses: “H_1_: cluster 1 represents a foreshock sequence” and “H_2_: cluster 2 represents a foreshock sequence.” Hypotheses H_1_ and H_2_ have been examined on the basis of the seismicity patterns discriminating foreshocks from swarms (Section 2.2).

#### 3.2.1. Testing H_1_

The seismicity states BGS 1 (Figure 7a) and cluster 1 (Figure 7b) are characterized by r = 0.17 events/day and r = 1.74 events/day, respectively (Figure 8). On the other hand, b_ML_ = 0.80 and b_GR_ = 0.89 were found for BGS 1, while increased values of b_ML_ = 1.26 and b_GR_ = 1.55 were found for cluster 1 (Figure 9a). The increases of both r and b are very highly significant (Table 1). To check the possible dependence of b on Μ_c_, the b_ML_ was re-calculated for M_c_ varying from 1.1 to 2.2 (Figure 9b). It is evident that b_ML_ in cluster 1 is systematically higher than that in BGS 1.

Cluster 1 was spatially concentrated mainly to the north and west of the mainshock epicenter. Before the onset of cluster 1, i.e., during the background seismicity state BGS 1, the mean distance <D> of epicenters from the mainshock epicenter ranged between 30 and 50 km (Figure 10). However, as soon as the cluster 1 appeared, <D> gradually dropped and ranged between 15 and 35 km. This trend has been verified by considering a variable number of events, w, ranging from 30 to 150, which implies that the observed drop of <D> is not biased. The statistical z-test showed that the drop of <D> is of very high significance (Table 1).

To check further the seismicity patterns characterizing cluster 1, the time variations of b_ML_ and <M> have been examined with the backward windowing technique detailed in Appendix B. It is observed that a gradual increase of b_ML_ and respective decrease of <M> already started from February 2016 (Figure 11). However, these changes became very strong only from the beginning of August 2016, which signifies the onset time of cluster 1. The changes continued until the end of the cluster by the beginning of May 2017.

Cluster 1 does not share the drop of b-value which characterizes foreshock sequences, although it exhibits significant increase in seismicity rate, r. Therefore, the hypothesis H_1_ that the cluster 1 represents a foreshock sequence is rejected. Instead, the increase of both r and b are features of transient seismic swarms that do not conclude with an event dominant in size. The gradual decrease of <D> suggests the movement of epicenters towards the mainshock epicenter and perhaps is related to the possible premonitory nature of the swarm (cluster 1), a point which is discussed later.

#### 3.2.2. Testing H_2_

The background seismicity BGS 2 (Figure 12a) was characterized by rate r = 0.18 events/day (white area in Figure 13) and b_ML_ = 1.23, b_GR_= 1.13 (Figure 14a). On the contrary, in cluster 2 (Figure 12b) the rate increased to r=0.54 events/day (gray area in Figure 13) but at the same time b decreased (b_ML_ = 0.88, b_GR_ = 0.71, Figure 14a). We found that for M_c_ varying from 1.1 to 2.2, the b-value remained constantly low (Figure 14b). Cluster 2 was spatially concentrated in the area to the north and northwest of the mainshock epicenter (Figure 12b). During the BGS 2 state, i.e., before the onset of cluster 2 on 20 April 2018, the average epicentral distance <D> ranged between 30 and 40 km (Figure 15). During the cluster 2, however, <D> gradually dropped and ranged between 20 and 30 km. The decrease of <D> is not biased since it has been verified for various values of w. We statistically tested and found that the <D> drop is of very high significance (Table 1). During BGS 2, a small non-significant cluster appeared in August-September 2017 (see Figure 13).

It is evident that a gradual decrease of b_ML_ and respective increase of <M> already happened from the beginning of March 2018 (Figure 16). However, these changes became very strong from 20 April 2018, which signifies the onset time of cluster 2. The changes continued until the generation of the 25 October 2018 mainshock.

The simultaneous increase of r and decrease of b-value indicate that cluster 2 shares the characteristic features of foreshocks, and therefore hypothesis H_2_ is verified. In addition, <D> decreases clearly during the foreshock sequence, which implies a move of foreshocks towards the mainshock epicenter. The three seismicity changes are highly significant (Table 1) and, therefore, the foreshock sequence starting on 20 April 2018 and culminating in the 2018 mainshock was a strong one.

### 3.3. Relocation

With the relocation of the entire seismic sequence, the average RMS from 0.4, as in the NOA catalogue, was reduced to 0.3, while the calculated average horizontal and vertical errors were found ± 2 km in three dimensions. The relocation results indicate that the seismicity in the entire time period examined occurred at shallow depths less than 20 km, which is generally consistent with the depths determined in the NOA catalogue. Of importance to our analysis is that very few events of the swarm and foreshock clusters were relocated outside the critical area of R = 30 km. As a consequence, the r, b and <D> changes remained at very high significance levels (Table 1). The relocated epicenters as well as the ones determined in the NOA catalogue for the 25 October 2018 mainshock, its two largest imminent foreshocks and the first 5-day aftershocks, are plotted in Figure 1b.

### 3.4. Rupture Process in Space and Time

The shallow depths (<20 km) of the seismicity associated with the 2018 mainshock along with the earthquake occurring above the megathrust domain suggests that the sequence likely ruptured the crust at the overriding plate which moves from N-NE to S-SW (Figure 1b). Fault strike of 10° and dip of 24° were found to better fit the data (Table 2). The heterogeneous temporal and spatial slip distribution in the fault area shows a complex rupture process. The time evolution of slip in the fault area is illustrated by six snaps using time intervals of nearly 3 s (Figure 17). The rupture process was fast enough given that the rupture velocity that better fit the data was found at 3.2 km/s. The total rupture duration was estimated close to 18 s. However, from the source time function (Figure 18), the results show that the release of seismic moment, M_o_, was rather complex. About half of M_o_ was released within the main slip patch during the first ~6 s of the rupture process, while the remaining moment release occurred with several peaks in the next 12 s of the process. The total seismic moment released was calculated at M_o_ = 2.0 × 10^26^ dyne*cm, which corresponds to magnitude M_w_ = 6.80.

Remarkable seismic slip exceeding 0.3 m was found at depths of more than ~7 km, but the largest co-seismic slip patch, which could be considered as an asperity, was confined at depths from 10 to 20 km (Figure 17). In the asperity, the maximum slip amplitude was of 1.8 m with the rake vector at that point being of 164°. As the rake was left to vary upon the fault, showing the type of rupture, an arithmetic mean for the cells with significant slip over 10 cm would give 140°. These values indicate oblique-slip motion with a dominating right-lateral strike-slip component but an important thrust-type component. The rupture was of a total length of ~70 km and propagated bilaterally but mainly southwards. At the north fault side, slip amplitudes as small as ~10 cm were found, taking the value of 20 cm only within a small patch.

It is noteworthy that the foreshocks area covered part of the swarm area, which delineates the locked patch (asperity) to the north-northeast (Figure 19). However, the foreshocks area was extended further southwards at locations up-dip of the asperity and partly within the upper side of it (Figure 19a, Appendix A). The early (first 5 days) aftershocks area (Figure 19b) is nearly coincident with the foreshocks area. These observations are of particular interest to better understand the mainshock nucleation and rupture process and, therefore, they are further examined in the Discussion section.

## 4. Discussion

Several mechanisms have been proposed to account for the earthquake nucleation and the generation of the various types of earthquake precursors and of seismic sequences. In the 1970s, two dominant models for earthquake forerunners were proposed [101]. In the dilatancy-diffusion model, e.g., [102], the earthquake occurs on a pre-existing fault and pore fluids play a central role. According to the dry model, e.g., [103], the earthquake occurs as a fault is formed; it thus involves fracture of intact rock on a scale comparable with the dimensions of the main rupture. Pore fluids are not required in the dry model. However, both models predict the occurrence of foreshocks shortly before the mainshock.

Laboratory observations and theoretical models both indicate that earthquake nucleation is accompanied by long intervals of accelerating slip and that earthquake activity can be modeled as a sequence of nucleation events [104]. Ιn this model, earthquake clustering, e.g., aftershocks, arises from sensitivity of nucleation times to the stress changes induced by prior earthquakes, while the model predicts two mechanisms for foreshocks. In the first, the stress change at a foreshock time increases the probability of earthquakes at all magnitudes, including the eventual mainshock. With the second model, accelerating fault slip on the mainshock nucleation zone triggers foreshocks. On the other hand, simulations of earthquake sequences showed [105] that inhomogeneity of the pre-stress and/or the static and dynamic frictional stresses combined with viscoelasticity of the medium provides a mechanism that accounts for the pre-seismic and the post-seismic creep and the stopping of the crack. It accounts also for the various earthquake sequence types that are observed in nature. The increase of the fluctuation amplitudes in the spatial distribution of stresses lead to a progressive variation of the earthquake sequence types from an “isolated earthquake,” to foreshocks, mainshock and aftershocks, and finally, to seismic swarms. The pore pressure variation, e.g., fluid injection, is a dominant model in the interpretation of seismic swarms, e.g., [106,107]. This mechanism is favored in the shallow parts of subduction zones since the high level of fluid pressure is a feature of such regions, e.g., [108].

Short-term foreshocks are observed in areas of various tectonic styles, e.g., [6,26], and precede mainshocks in a timeframe which as a rule does not exceed ~6 months. On the other hand, premonitory intermediate-term and long-term swarms have been recognized long ago in various seismogenic regions [109,110], including Greece [108]. This has been interpreted as meaning that the swarms are part of a long-term process which culminates in the major earthquake [111]. There is evidence that foreshocks and swarms control the rupture process of the mainshock at least in subduction plate boundaries, e.g., [112,113]. The 1 April 2014, Northern Chile M_w_=8.1 Iquique earthquake is a well-studied case, although contradictory results have been found regarding the timeframe and spatial extent of the foreshock sequence [60,113,114,115,116,117,118,119,120,121].

In the case examined here, after a period of steady-state background seismicity, the low-slip intermediate-term premonitory seismic swarm started about 2.3 years before the mainshock, lasted for ~10 months and concentrated just to the north of the asperity area (Figure 17). The swarm activity ceased and turned to the background seismicity state for ~1 yr. Then, in the last six months prior the mainshock, the activity resumed as an accelerated foreshock sequence covering partly the swarm area to the north of the asperity. That the foreshocks area extended further to south up-dip of the asperity area and partly within the upper side of it (Figure 19a) suggests that foreshocks possibly contributed to stress accumulation in the locked patch before the 2018 M_w_ = 6.8 mainshock. The main event nucleated near the southern deep end of the foreshocks area very close to the hypocenters of the two largest imminent foreshocks occurring 8 days (M_w_ = 4.1) and 50 min (M_w_ = 4.8) prior the mainshock (Figure 1b). This is suggestive of the critical role that foreshocks played in unlocking the asperity, thus resembling the mechanism proposed for the M_w_ = 8.1 Iquique earthquake in the Chilean subduction zone [115,118,119,120].

During the first ~6 s, the dynamic rupture front propagated for ~10 km from the hypocenter towards N-NE and down-dip (Figure 17). However, the northwards rupture propagation stopped soon at the boundary of the foreshocks and swarm area. Then, the rupture turned southwards where it continued for about 9 s. In about the last 6 s of the process, a small slip patch appeared northwards but beyond the swarm/foreshocks area (Figure 17). However, the remaining area between the main and small patches hosted very little slip. Only a small number of aftershocks occurred in the asperity and in the southwards extension of the rupture. We found that most of foreshocks and early aftershocks are located mainly in the surrounding area of the asperity (Figure 19), which is a reminiscent of other cases, e.g., [3,112]. It suggests that the asperity was a barrier during the swarm and foreshocks periods. After the mainshock, the stress in the area surrounding the asperity likely increased and triggered most of early aftershocks.

To test further this suggestion, we performed Coulomb Stress Change (CSC) modeling which has been extensively used as physics-based tool for aftershock forecasts, e.g., [122,123]. CSCs were calculated based on the complex slip distribution of the Zakynthos mainshock rupture at several depths (5, 10, 13, 15 km) and using the code Strop [124] in an elastic half-space [125]. We assumed shear modulus of 3.0 × 10^10^ Pa, Poisson’s ratio 0.25 and effective coefficient of friction μ’ = 0.4. The results obtained are quite similar for all depths. Figure 20 shows the CSC at a depth of 13 km. It is evident that early aftershocks occurred not only in positive but also in negative CSC lobes, thus occupying the swarm/foreshock area. Errors in the relocation or in the slip distribution process could yield an explanation for the aftershocks distribution. However, since most aftershocks have small source depths and taking into account that the mainshock fault dips towards the east, we suggest that aftershocks distribution very likely has been controlled not only by the stress released by the mainshock rupture process but also by pre-existing stress in the swarm/foreshock area.

We suggest that the intermediate-term swarm occurred in a zone of the fault possibly of very low moment release. The swarm activity possibly triggered the short-term foreshock sequence which expanded further southwards and up-dip of the mainshock asperity. In the foreshocks area, the stresses were continuously relaxed, but the main rupture initially failed to break out. However, the accelerated progress of foreshocks may have gradually promoted unlocking of the seismic fault during the nucleation phase. On the other hand, the precursory nature of the strong foreshock sequence, expressed by the characteristic patterns of increase in r and decrease in b and <D>, is quite clear and verifies the patterns found in other foreshock sequences.

It has been suggested that early identification of foreshock precursory patterns in quasi real-time conditions based on regular seismicity monitoring may help for predicting the mainshock, e.g., Ref. [16]. The northwestern Hellenic Subduction Zone certainly would be a target area for such an experiment for the reason that it has been repeatedly ruptured by strong shallow earthquakes before 2018, the most recent ones being those of 11 May 1976 (M_w_ = 6.43, 37.370° N, 20.345° E) and 18 November 1997 (M_w_ = 6.58, 37.509° N, 20.844° E) [126] (Figure 1b). Within a program of regular seismicity analysis for precursory patterns identification, the months-long organization of a spatiotemporal seismicity cluster hardly could escape the attention of staff in institutes, such as NOA, charged with daily seismograph monitoring. Then, the ongoing foreshock sequence, i.e., the cluster 2 analyzed here, would have been recognizable by around 1 July 2018, if not earlier, since by that date the foreshock sequence organization was already well-shaped and detectable. Retrospective testing showed that by that date the seismicity changes already passed the 95% significance level.

One may argue that a near real-time analysis could not be performed since the epicenter of the forthcoming mainshock was unknown beforehand. However, this difficulty would be easily overcome if one reasonably considered a target area geographically determined by the epicenters of past strong mainshocks, e.g., the 1976 and 1997 ones, as well as by a few, say three, additional nearby hypothetical epicenters. Then, a real-time seismicity analysis with the statistical methodology explained in this study certainly would reveal the ongoing foreshock activity. A remaining puzzling issue is the assessment of magnitude, M, of the forthcoming mainshock since no relation has been found so far between M and parameters of the foreshock sequence, such as the largest foreshock magnitude and the sequence duration. Preliminary results, however, show that M perhaps scales with the foreshocks area for a wide range of M [19]. Magnitude M ranging from 6.5 to 7 fits quite well with this preliminary result for a foreshocks area of ~30–40 km that would have been detected by 1 July 2018.

## 5. Conclusions

Foreshock sequences have important predictive value since their precursory nature is recognizable from the space-time-size patterns discriminating them from the background seismicity and other seismicity clusters such as swarms and aftershocks. During foreshock sequences, the seismicity rate (event count) increases in a timeframe of usually less than 6 months, the b-value drops while the events usually move towards the mainshock epicenter. These patterns have been verified with the case of Zakynthos 25 October 2018 M_w_ = 6.8 shallow (h = 13 km) mainshock. The mainshock was preceded by two highly significant seismicity clusters: an intermediate-term swarm (2.3–1.5 yrs prior the mainshock) and a short-term foreshock sequence (last 6 months prior the mainshock). Both clusters developed to the north of the mainshock rupture, but the foreshock activity extended also southwards and up-dip of the mainshock asperity. The earthquake ruptured possibly within the overriding plate which moves from N-NE to S-SW at the NW sector of the Hellenic Subduction Zone. The mode of motion has been oblique-slip (strike-slip/thrust) with fault strike 10°, dip 24°, maximum length ~70 km and depth of faulting 24 km. The maximum slip amplitude in the asperity was 1.8 m. The rupture has been of total duration ~18 s and velocity ~3.2 km/s. The total seismic moment released was calculated at 2.0 × 10^26^ dyne*cm. Half of the moment released within the main slip patch (asperity) during the first ~6 s of the rupture process, while the remaining moment released with several peaks in the rest 12 s of the process.

The rupture propagated bilaterally but mainly southwards since to the north the rupture was bounded by the swarm/foreshocks area and, therefore, after the first 6 s of slip it turned southwards. The asperity ruptured down-dip of the foreshocks area. Part of the foreshock activity, including the two largest imminent foreshocks of M_w_ = 4.2 and M_w_ = 4.8, occurred just at the upper side of the asperity, very close to the mainshock hypocenter. Very likely the accelerated foreshock sequence promoted slip accumulation that contributed to unlock the asperity and break with the mainshock. Most foreshocks and early aftershocks have been located mainly in the surrounding area of the asperity, indicating that the asperity behaved as a barrier during the foreshocks period. The stress in the area surrounding the asperity very likely increased after the mainshock and triggered early aftershocks. However, pre-existing stress in the swarm and foreshock area may have played role in the aftershocks generation.

## Figures and Tables

**Figure 1 sensors-20-05681-f001:**
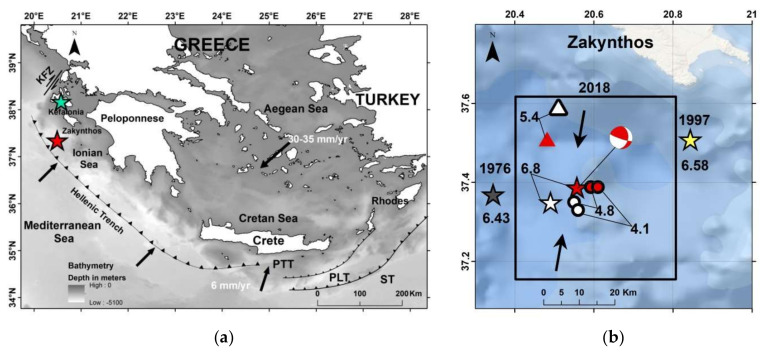
Elements of the Hellenic Subduction Zone (HSZ) (**a**); arrows and nearby figures show directions and velocities of lithospheric plates motion, respectively; toothed lines remark tectonic trenches along the HSZ; PTT = Ptolemy trench, PT = Pliny trench; ST = Strabo trench, KFZ = Kefalonia Fault Zone. Red and green stars illustrate relocated epicenters of the 25 October 2018 M_w_ = 6.8 earthquake (this paper) and the 26 January 2014 M_w_ = 6.0 earthquake (see text). Main events of the 2018 earthquake sequence and of past seismicity (**b**); within the inset are epicenters of the 25 October 2018 mainshock (star), of the largest imminent 17 and 25 October 2018 foreshocks (circles) and of the largest 30 October 2018 aftershock (triangle). Open symbols are determinations from the National Observatory of Athens (NOA) catalogue, red symbols are epicenters relocated in this paper. Figures near epicenters show moment magnitudes. Epicenters and magnitudes of the 11 May 1976 and 18 November 1997 mainshocks are determined by ISC-GEM Global Instrumental Earthquake Catalogue (http://doi.org/10.31905/D808B825). Beach-ball is the Global Centroid Moment Tensor Project fault-plane solution (https://www.globalcmt.org/CMTsearch.html) and arrows show inferred plate motion.

**Figure 2 sensors-20-05681-f002:**
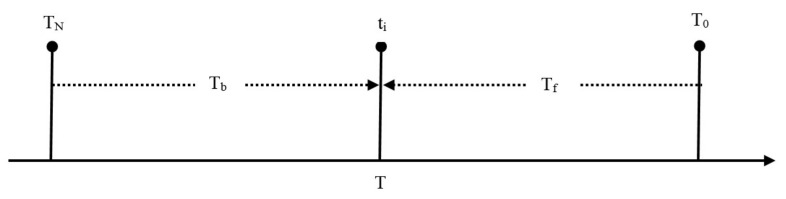
Schematic diagram for seismicity changes identification. A time segment T of the earthquake catalogue is divided in sub-intervals with supposedly different seismicity properties: e.g., T_b_ is background seismicity, T_f_ is a cluster, e.g., foreshock sequence, prior the mainshock origin time, T_0_. Suppose T_f_ is characterized by significant seismicity changes with respect to T_b_. To identify seismicity changes, a long number of T_b_/T_f_ pairs are examined by shifting time t_i_ forwards or backwards. In practice, the time lengths of T_b_ and T_f_ are changed by removing one time unit (e.g., 1 day) or a certain number of events from T_b_ and adding to T_f_ inversely. The start and end times of the changes are determined from the maximum significance levels found among all possible pairs (Section 2.2).

**Figure 3 sensors-20-05681-f003:**
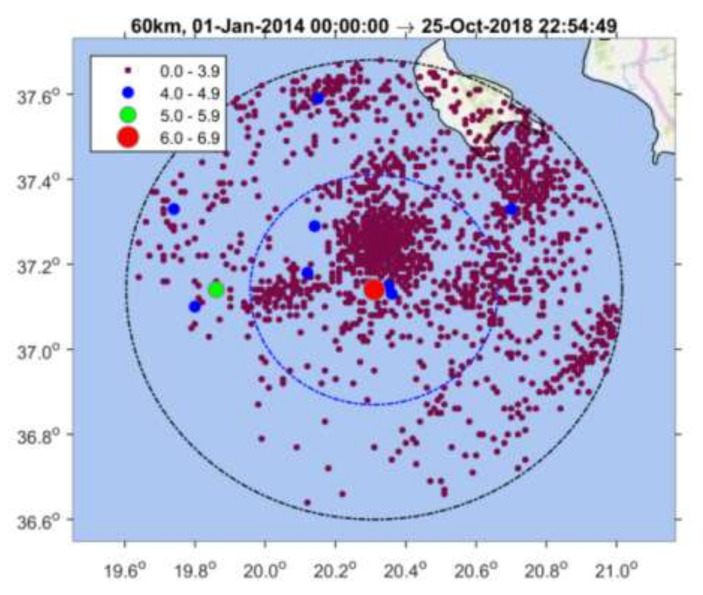
Spatial distribution of earthquakes listed in the National Observatory of Athens (NOA) catalogue and occurring from 1 January 2014 to the 25 October 2018 mainshock occurrence (red circle) within radii of R = 60 km and R = 30 km from the mainshock epicenter. Color panel shows scaling of earthquake magnitude size.

**Figure 4 sensors-20-05681-f004:**
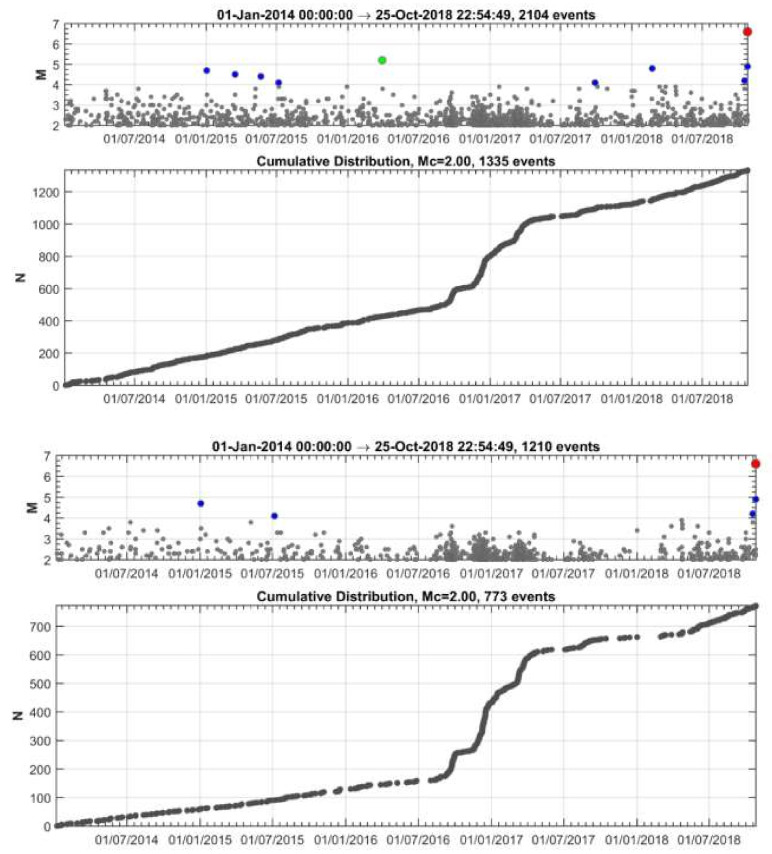
Time distribution of earthquakes listed in NOA catalogue and occurring from 1 January 2014 to the 25 October 2018 mainshock occurrence (red circle) within radii of R = 60 km (upper panel) and R = 30 km (lower panel) from the mainshock epicenter. In each panel all the catalogued events as well as the cumulative number, N, of only events of M_c_ ≥2.0 are plotted. N versus time shows four seismicity stages: (1) 1 January 2014–1 August 2016 (background seismicity BGS 1), (2) 1 August 2016–1 May 2017 (cluster 1), (3) 1 May 2017–20 April 2018 (background seismicity BGS 2), (4) 20 April 2018–25 October 2018 (cluster 2). Colored dots shows earthquake size as in Figure 3. Only earthquakes of M_L_ < 4 occurred during the cluster 1 (swarm). Two imminent foreshocks (M_w_ = 4.1, M_w_ = 4.8, Figure 1b) occurred during the foreshock sequence (cluster 2).

**Figure 5 sensors-20-05681-f005:**
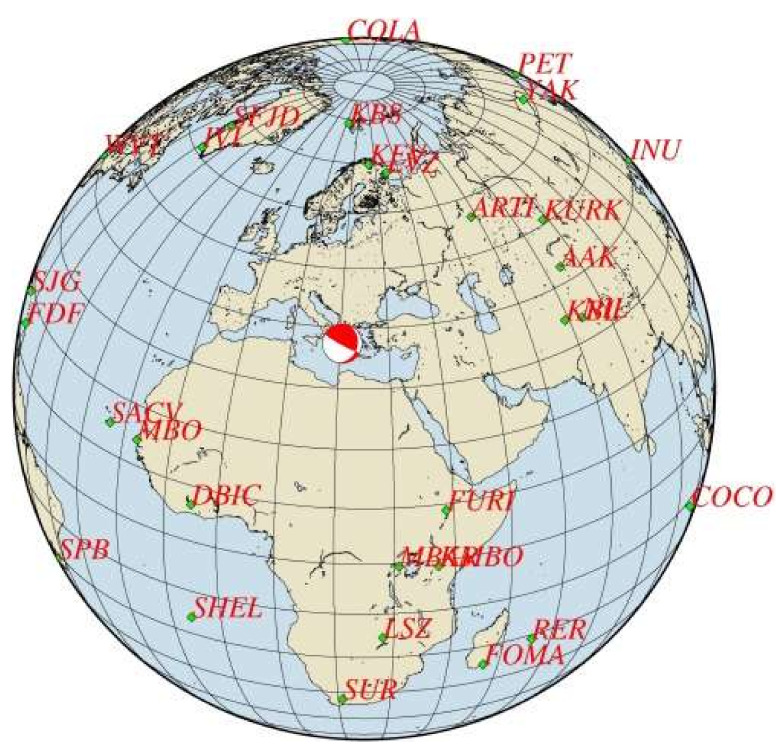
P-wave records of the 25 October 2018 mainshock were downloaded from 30 teleseismic stations at epicentral distances 30° < Δ <90° beach ball as in Figure 1b, and also shows the mainshock area.

**Figure 6 sensors-20-05681-f006:**
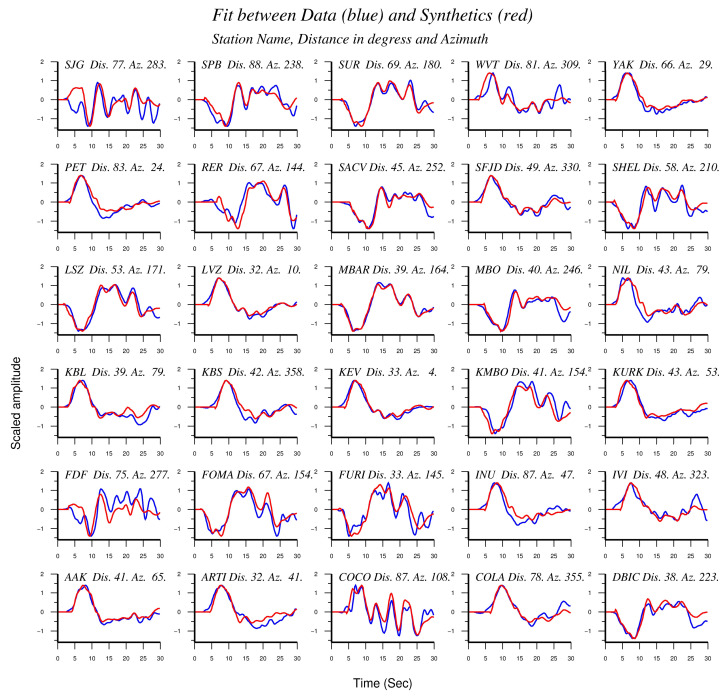
Fit between real P-waveforms and synthetics for the 30 stations illustrated in Figure 5.

**Figure 7 sensors-20-05681-f007:**
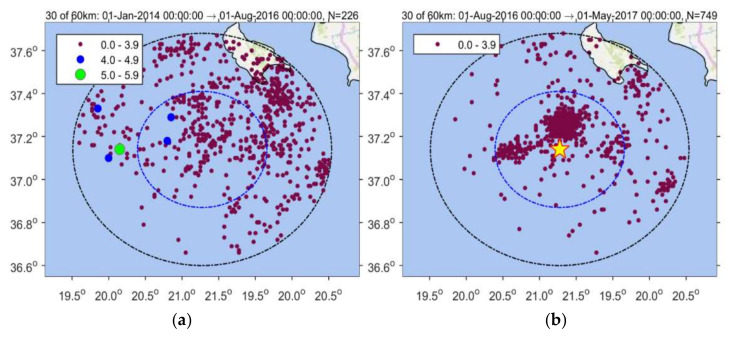
Seismicity within radii of 30 km and 60 km around the 25 October 2018 mainshock epicenter from 1 January 2014 up to 1 August 2016 (background seismicity state BGS 1) (**a**), and from 1 August 2016 up to 1 May 2017 (cluster 1) (**b**). Color panel illustrates earthquake scaling of magnitude size; star shows mainshock epicenter.

**Figure 8 sensors-20-05681-f008:**
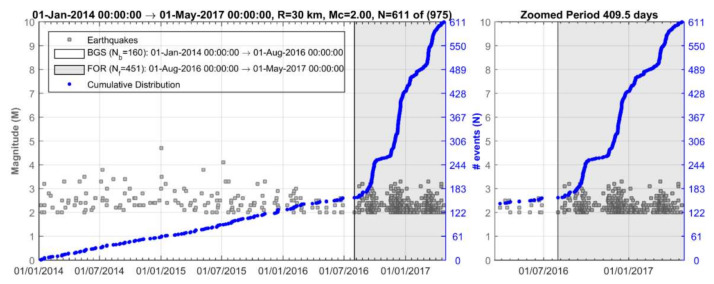
Time distribution of earthquakes (for M_c_ ≥ 2.0 number of events N = 611 out of 975 totally listed in NOA catalogue) occurring during the background seismicity state (BGS 1, 1 January 2014–1 August 2016, number of events N_b_) and the hypothetical foreshock activity (FOR-cluster 1, grey-shaded area, 1 August 2016–1 May 2017, number of events N_f_). Cluster 1 is composed of three sequential sub-clusters. Plot in blue is the cumulative number of events versus time. Seismicity testing showed that cluster 1 has not been a foreshock sequence but a transient swarm activity (see Section 3.2.1). The plot in grey area is zoomed from 1 March 2016.

**Figure 9 sensors-20-05681-f009:**
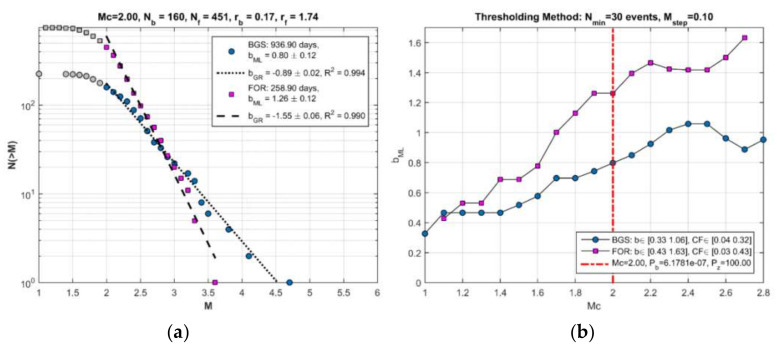
Magnitude (M)-frequency (N) diagram of the seismicity state BGS 1 (number of events N_b_, seismicity rate r_b_, correlation coefficient R) and of the hypothetical foreshock sequence (FOR-cluster 1; N_f_, r_f_, as in Figure 8) (**a**); b_ML_ and b_GR_ are b-value estimates from the maximum likelihood approximation and the weighted least-square method, respectively. The b-value in BGS 1 is significantly lower than that in FOR-cluster 1 period. This is systematically valid for M_c_ varying from 1.1 to 2.7 with step 0.1 (**b**). For M_c_ ≥ 2.0 (vertical red line) the changes in both b and r are highly significant (see Section 3.2.1). Seismicity testing showed that the hypothetical FOR period (cluster 1) does not really represent foreshock activity but it has been a transient swarm activity (see Section 3.2.1).

**Figure 10 sensors-20-05681-f010:**
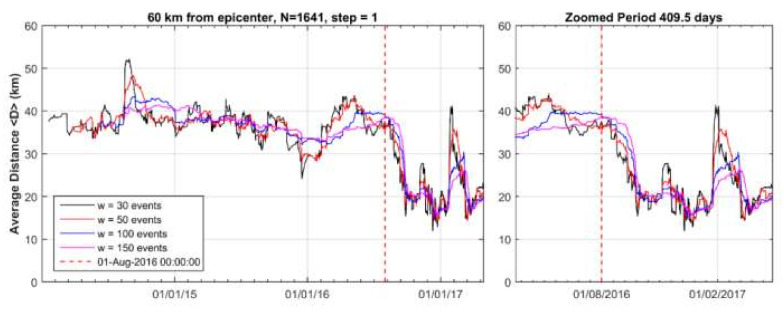
Time variation of the average distance, <D>, between the 2018 mainshock epicenter and earthquake (M_c_ ≥ 2.0) epicenters during the seismicity stage BGS 1 (1 January 2014–1 August 2016) and the hypothetical foreshock activity (cluster 1, 1 August 2016–1 May 2017) within radius R=60 km from the 2018 mainshock epicenter. In each step, <D> is calculated as the mean of w events. After 1 August 2016 (vertical red line) <D> dropped significantly (see Section 2.3). Two peaks around November 2016 and March 2017 correspond to sub-clusters shown in Figure 8.

**Figure 11 sensors-20-05681-f011:**
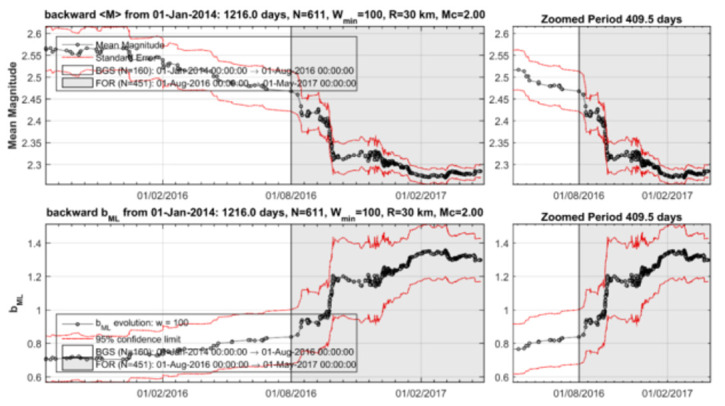
Time variation of the mean magnitude <M> (upper panel) and b (lower panel) for M_c_ ≥ 2.0 during the seismicity stage BGS 1 and the hypothetical foreshock activity (cluster 1) within radius R=30 km from the 2018 mainshock epicenter.

**Figure 12 sensors-20-05681-f012:**
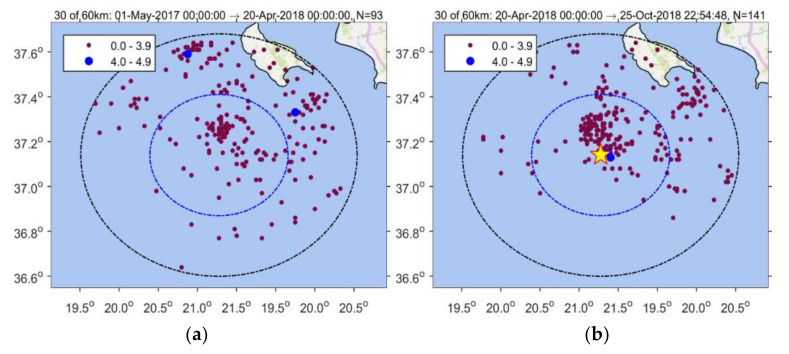
Seismicity at radii of 30 km and 60 km around the 25 October 2018 mainshock epicenter from 1 May 2017 up to 20 April 2018 (background seismicity BGS 2) (**a**), and from 20 April 2018 up to 25 October 2018 just prior to the mainshock occurrence (cluster 1) (**b**). Color panel shows scaling of earthquake magnitude size; star is mainshock epicenter.

**Figure 13 sensors-20-05681-f013:**
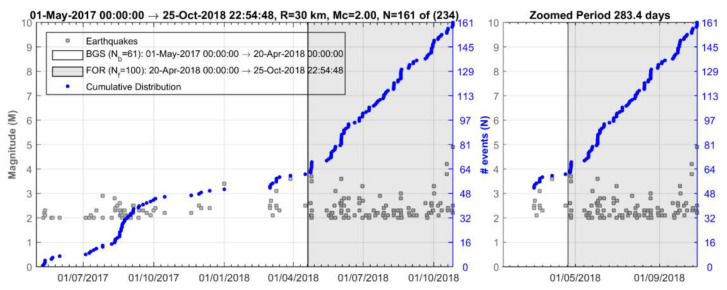
Time distribution of earthquakes (for Mc ≥ 2.0 number of events N=161 out of 234 totally listed in the NOA catalogue) occurring during the 1 May 2017–20 April 2018 background seismicity period BGS 2 (number of events N_b_) and the hypothetical foreshock activity (FOR-cluster 2, grey-shaded area, 20 April 2018–25 October 2018, number of events N_f_). Plot in blue is the cumulative number of events versus time. Vertical bar shows a date of 20 April 2018. Τhe same plot is zoomed from 1 February 2018 onwards. Seismicity testing showed that cluster 2 was a typical foreshock sequence (see Section 3.2.1). Two imminent foreshocks (M_w_ = 4.1, M_w_ = 4.8; see magnitude scale in gray area, also Figure 1b) were the largest ones during the entire foreshock sequence.

**Figure 14 sensors-20-05681-f014:**
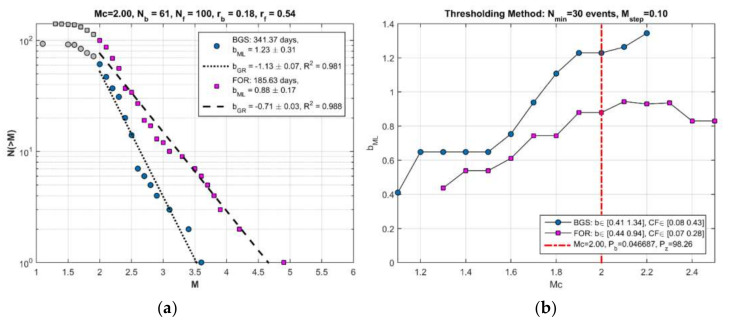
As in Figure 9 for the periods of background seismicity BGS 2 and FOR-cluster 2 (**a**). The b-value in BGS 2 is significantly higher than that in FOR period. This is systematically valid for M_c_ varying from 1.1 to 2.2 with step 0.1 (**b**). For M_c_ = 2.0 (vertical red line) the differences in b are higlhy significant (see Section 3.2.1). Seismicity testing showed that the FOR period (cluster 2) represents a typical foreshock sequence (see Section 3.2.1).

**Figure 15 sensors-20-05681-f015:**
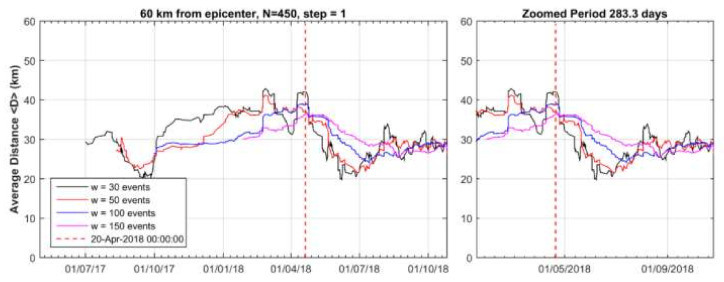
Variation of <D> for the BGS 2 background seismicity state (1 May 2017–20 April 2018) and the hypothetical foreshock activity (20 April 2018–25 October 2018, cluster 2). In each step <D> is calculated as the mean of w events. After 20 April 2018 (vertical red line) <D> dropped significantly with respect to the previous period (see Section 2.3).

**Figure 16 sensors-20-05681-f016:**
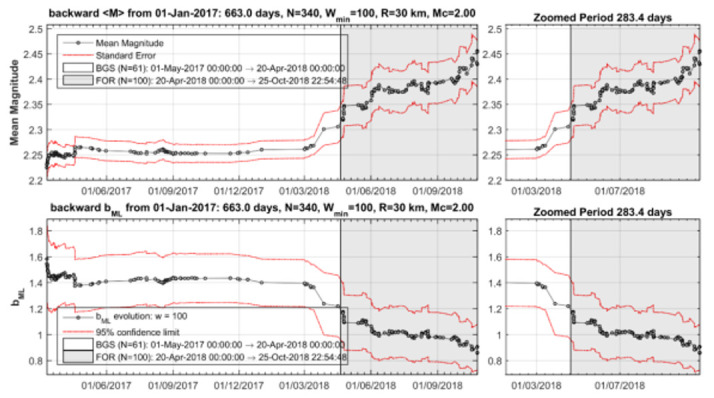
Variation of <M> (upper panel) and b_ML_ (lower panel) for the seismicity plotted in Figure 13. After the 1 May 2017–20 April 2018 background seismicity state (BGS 2), <M> increased while b_ML_ decreased systematically during the foreshock activity (cluster 2-FOR, grey-shaded area, 20 April 2018–25 October 2018).

**Figure 17 sensors-20-05681-f017:**
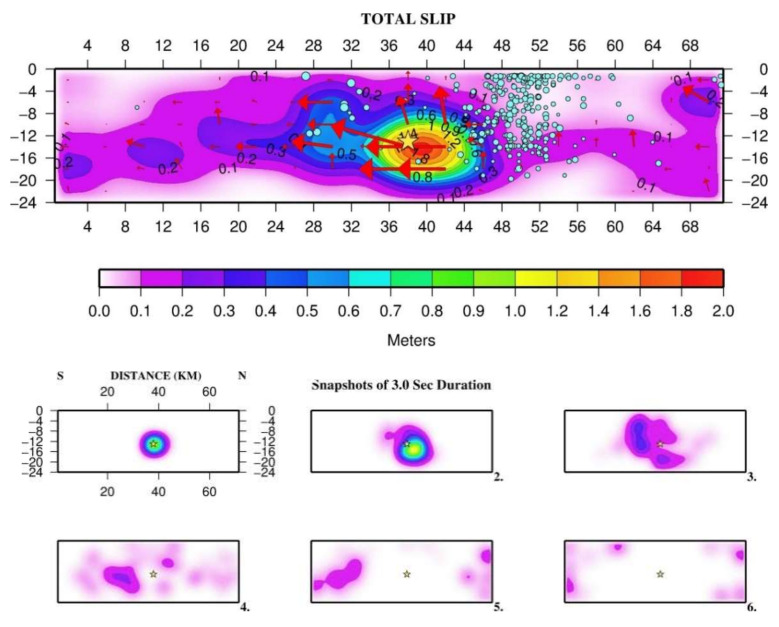
Space (upper panel) and space-time (lower panel in snapshots of 3 s in duration) variation of the co-seismic slip during the 25 October 2018 mainshock (section from south to north); star is relocated epicenter. Red arrows show slip direction, length of arrows scales with amount of slip (black figures in m). Blue circles are relocated earthquakes in the premonitory intermediate-term (1 August 2016–1 May 2017) transient swarm. The swarm area delineates the asperity to the north. Within the first 6 s the rupture propagated northwards but stopped in the boundary with the swarm area.

**Figure 18 sensors-20-05681-f018:**
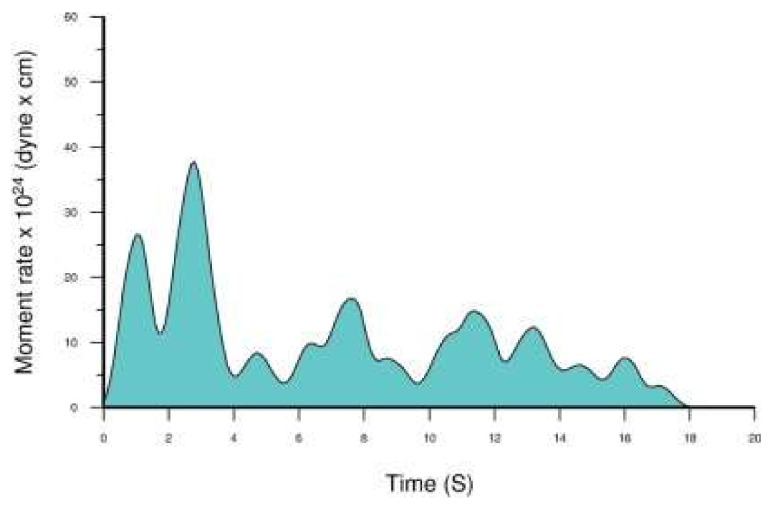
Source time function of the 25 October 2018 mainshock.

**Figure 19 sensors-20-05681-f019:**
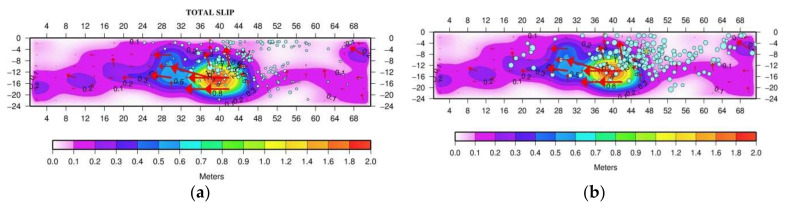
As in Figure 17, blue circles show relocated foreshocks (20 April 2018–25 October 2018) (**a**) and first 5-day aftershocks (**b**). Star outside the main slip (asperity) area and to the north of it is the epicenter of the largest aftershock (30 October 2018, M_w_ = 5.4, Figure 1b) of the entire aftershock period up to September 2020. Foreshocks delineated the asperity both to the north and up-dip. The foreshocks area overlaps partly with the upper side of the asperity. The foreshocks and aftershock areas are nearly identical.

**Figure 20 sensors-20-05681-f020:**
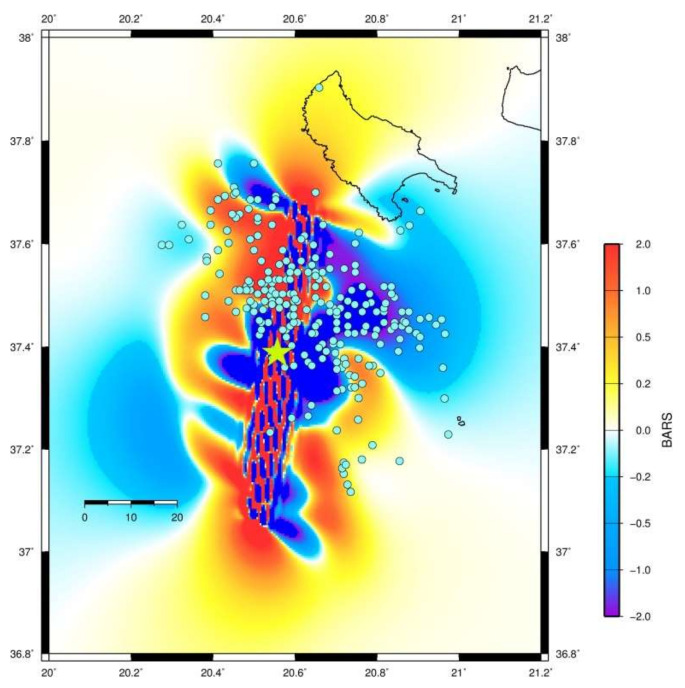
Coulomb Stress Change modeling for the mainshock of 25 October 2018 (star).

**Table 1 sensors-20-05681-t001:** Results of testing for the b, r, and <D> changes within radius of R = 30 km from the 25 October 2018 mainshock epicenter. Lower magnitude threshold is M_c_ = 2. Key: BGS = background seismicity; n = number of events; P = probability for the significance level of b-value change according to Utsu-test; P_z_ = probability according to z-test.

Seismicity Period	Start Date h:m:s	End Date h:m:s	n	P	1-P*_z_* (for *r*)	1-P*_z_* (for *D*)
BGS 1	01 Jan 2014 00:00:00	01 Aug 2016 00:00:00	160		100	100
Cluster 1	01 Aug 2016 00:00:01	01 May 2017 00:00:00	451	6.18 × 10^-7^		
BGS 2	01 May 2017 00:00:01	20 Apr 2018 00:00:00	61		98.26	99.97
Cluster 2	20 Apr 2018 00:00:01	25 Oct 2018 22:54:48	100	0.0467		
Relocated BGS 2	01 May 2017 00:00:01	20 Apr 2018 00:00:00	58		98.93	100
Relocated Cluster 2	20 Apr 2018 00:00:01	25 Oct 2018 22:54:48	94	0.032		

**Table 2 sensors-20-05681-t002:** Best fit model parameters for the fault plane. L: fault length, H: depth of faulting, v: rupture velocity, h: focal depth, M_o_: seismic moment (dyne*cm), M_w_: moment magnitude. Dip direction of the fault is to the east, rake is calculated at the main slip patch of the fault (see text).

Strike	Dip	Rake	L (km)	H (km)	v (km/s)	h (km)	M_o_	M_w_
10°	24°	164°	70	24	3.2	13	2.0 × 10^26^	6.80

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
