# Peer review of "Short-Term Foreshocks as Key Information for Mainshock Timing and Rupture: The Mw6.8 25 October 2018 Zakynthos Earthquake, Hellenic Subduction Zone"

_sensors, 2020, doi:10.3390/s20195681_

Round 1
Reviewer 1 Report
Please see attached file.

Author Response
Reply to Review 1
Major comments:
- How did you compute seismicity rate r?
Least-squares method.
- Table 1, for Utsu test of b value, Z test of r and D, it might be more reasonable to take BGS1 as a reference background and test the parameters in Cluster 1, BGS2, and Cluster2 against it, so that one could see the temporal evolution of theses parameters during the earthquake preparation process.
This is a good idea but each period of anomalous seismicity is compared with the immediate preceding background seismicity period.
- For Z test, the significance level is more commonly used. To avoid misunderstanding, better to give 1-Pz.
Done in Table 1.
- Line 222, better to give the equation of AIC changes (dA).
The equation and relevant explanations have been inserted in the Appendix A.
- As seen from equation (3), b value is determined by if Mc is fixed in the analysis. Therefore, there is no need to show variations of in Figures 9 and 13. Instead, present the variations of r.
In principle the reviewer is right, we have explained this issue better in the text, lines 232-235.
- Line5 612-615. I cannot agree with the authors on this. The spatial distribution of early aftershocks may be controlled by the stress change produced by main shock, not by foreshocks.
This comment is practically the same with the next (n. 7).
- Line 665-666. As the authors claimed that the stress in the area surrounding the asperity possibly increased after mainshock and triggered most of the aftershocks, it might be useful to compute the Coulomb Failure Stress change (â–³ CFS) and check if the early aftershocks locate in the positive â–³CFS zone.
In the Discussion section of the revised version we modelled stress changes, explained the technique and software used, illustrated the results in a relevant figure (figure 20) and discussed the results.
Minor comments:
- The quality of figures should improve. Such as, Figures 5,9,13.
Improved.
- Figure 14, should give color bar.
Done.
- Line 64, “understand” to “understanding”.
Done.
- Line 102, “2020” to “2018”.
Done.
- Line 526, “7 m” to “7 km”.
Done.
- Line 545, Yellow star is not clear in the figure.
A biger star symbol was inserted.
Reviewer 2 Report
Figures 3 and 4 are easier to compare when side by side. Figures 3a and 4a should be done with the same resolution.
Figures 9b are not readable. They should be redone.
The article is written very clearly. Perhaps the authors should have cited the article
Mjachkin, V. I., Brace, W. F., Sobolev, G. A., & Dieterich, J. H. (1975). Two models for earthquake forerunners. In Earthquake Prediction and Rock Mechanics (pp. 169-181). Birkhäuser, Basel.
Author Response
Reply to Review 2
Figures 3 and 4 are easier to compare when side by side. Figures 3a and 4a should be done with the same resolution.
Figures improved.
Figures 9b are not readable. They should be redone.
Figures improved.
The article is written very clearly. Perhaps the authors should have cited the article
Mjachkin, V. I., Brace, W. F., Sobolev, G. A., & Dieterich, J. H. (1975). Two models for earthquake forerunners. In Earthquake Prediction and Rock Mechanics (pp. 169-181). Birkhäuser, Basel.
This article is cited and discussed in the Discussion section.
Reviewer 3 Report
This paper investigates in detail the foreshocks of Mw6.8 earthquake, and is worthy of being published with minor revision.一
Comments:
1, the foreshocks should give some key information about mainshock timing and rupture, not “control” .
2, The figures 3b, figure 8a, figure 8b, figure 9 should be improved, for instance text in these figures are too small, and the figures are not so clear.
Author Response
Reply to Review 3
This paper investigates in detail the foreshocks of Mw6.8 earthquake, and is worthy of being published with minor revision.一
Comments:
1, the foreshocks should give some key information about mainshock timing and rupture, not “control”.
A relevant correction in the paper title has been made.
2, The figures 3b, figure 8a, figure 8b, figure 9 should be improved, for instance text in these figures are too small, and the figures are not so clear.
Figures improved.
Reviewer 4 Report
I have found this paper by Papadopoulos et al really interesting. I think it deserves a quick publication after some minor revisions. In particular Authors should clear (at least infer) why
observable precursors are observable in the studied area. How many similar tectonic contexts exists in the world? Where are they located? What about further geophysical parameters (e.g.GPS) during the observation period?
Author Response
Reply to Review 4
I have found this paper by Papadopoulos et al really interesting. I think it deserves a quick publication after some minor revisions.
In particular Authors should clear (at least infer) why observable precursors are observable in the studied area.
How many similar tectonic contexts exists in the world?
Where are they located?
What about further geophysical parameters (e.g.GPS) during the observation period?
We appreciate it very much these questions by the reviewer. Our response is the next.
In the text of the paper, 3rd paragraph of Discussion section, we cited several papers referring to similar seismicity clusters observed in various regions of the world. In addition, we mention particularly the 1 April 2014, Northern Chile Mw=8.1 Iquique earthquake since it has been supported that foreshock activity unlocked the mainshock asperity as we support in our case.
As regards GPS observations, in Section “2. Materials and Methods”, subsection “The 2018 earthquake”, we cite several papers published about the 2018 mainshock and explain that in some of these papers co-seismic GPS measurements were inverted for modeling of the earthquake source. However, no GPS or other precursory observations are known to us before that mainshock.